# Brainstem Quadruple Aberrant Hyperphosphorylated Tau, Beta-Amyloid, Alpha-Synuclein and TDP-43 Pathology, Stress and Sleep Behavior Disorders

**DOI:** 10.3390/ijerph18136689

**Published:** 2021-06-22

**Authors:** Lilian Calderón-Garcidueñas, Ravi Philip Rajkumar, Elijah W. Stommel, Randy Kulesza, Yusra Mansour, Adriana Rico-Villanueva, Jorge Orlando Flores-Vázquez, Rafael Brito-Aguilar, Silvia Ramírez-Sánchez, Griselda García-Alonso, Diana A. Chávez-Franco, Samuel C. Luévano-Castro, Edgar García-Rojas, Paula Revueltas-Ficachi, Rodolfo Villarreal-Ríos, Partha S. Mukherjee

**Affiliations:** 1Department of Biomedical & Pharmaceutical Sciences, College of Health, The University of Montana, Missoula, MT 59812, USA; 2Universidad del Valle de México, Mexico City 14370, Mexico; adriana.rico@laureate.mx (A.R.-V.); jorge.flores@laureate.mx (J.O.F.-V.); rbritomu@hotmail.com (R.B.-A.); silvia.ramirezsa@uvmnet.edu (S.R.-S.); griselda.garcia@uvmnet.edu (G.G.-A.); alejeje01@gmail.com (D.A.C.-F.); y2j.luevano@hotmail.com (S.C.L.-C.); edgar.garcia@uvmnet.edu (E.G.-R.); paulirev97@gmail.com (P.R.-F.); 3Department of Psychiatry, Jawaharlal Institute of Postgraduate Medical Education and Research, Pondicherry 605006, India; ravi.psych@gmail.com; 4Department of Neurology, Geisel School of Medicine at Dartmouth, Hanover, NH 03755, USA; Elijah.W.Stommel@hitchcock.org; 5Auditory Research Center, Lake Erie College of Osteopathic Medicine, Erie, PA 16509, USA; rjkulesza@gmail.com; 6Henry Ford Macomb, Department of Otolaryngology—Facial Plastic Surgery, Lake Erie College of Osteopathic Medicine, Erie, PA 16509, USA; Ymansour49902@med.LECOM.edu; 7Universidad Autónoma de Piedras Negras, Piedras Negras 26000, Mexico; rvillarreal45@hotmail.com; 8Interdisciplinary Statistical Research Unit, Indian Statistical Institute, Kolkata 700108, India; psmukherjee.statistics@gmail.com

**Keywords:** Alzheimer, dementia, quadruple proteinopathies, nanoparticles, post-traumatic stress disorder, PTSD, possible REM sleep behavior disorder, pRBD, TDP-43 proteinopathies, Parkinson, Mexico

## Abstract

Quadruple aberrant hyperphosphorylated tau (p-τ), amyloid-β peptide, alpha-synuclein and TDP-43 brainstem and supratentorial pathology are documented in forensic ≤40y autopsies in Metropolitan Mexico City (MMC), and p-τ is the major aberrant protein. Post-traumatic stress disorder (PTSD) is associated with an elevated risk of subsequent dementia, and rapid eye movement sleep behavior disorder (RBD) is documented in PD, AD, Lewy body dementia and ALS. This study aimed to identify an association between PTSD and potential pRBD in Mexico. An anonymous online survey of 4502 urban college-educated adults, 29.3 ± 10.3 years; MMC, n = 1865; non-MMC, n = 2637, measured PTSD symptoms using the Impact of Event Scale–Revised (IES-R) and pRBD symptoms using the RBD Single-Question. Over 50% of the participants had IES-R scores ≥33 indicating probable PTSD. pRBD was identified in 22.6% of the participants across Mexico and 32.7% in MMC residents with PTSD. MMC subjects with PTSD had an OR 2.6218 [2.5348, 2.7117] of answering yes to the pRBD. PTSD and pRBD were more common in women. This study showed an association between PTSD and pRBD, strengthening the possibility of a connection with misfolded proteinopathies in young urbanites. We need to confirm the RBD diagnosis using an overnight polysomnogram. Mexican women are at high risk for stress and sleep disorders.

## 1. Introduction

Recent evidence from around the world has identified a significant role for air pollution in the pathophysiology of dementia [1,2,3,4,5]. In our earlier research, neuropathological hallmarks of Alzheimer’s disease (AD), Parkinson’s disease (PD) and transactive response DNA-binding protein TDP-43 pathology in young megacity residents have been documented at post-mortem, even in toddlers [6,7,8,9,10]. These changes are associated with specific forms of air pollution, including exogenous magnetic Fe-rich nanoparticles resulting from fuel combustion and engineered Titanium (Ti) nanorods from e-waste [6,9,11]. In keeping with the quadruple abnormal protein pathology [6], Metropolitan Mexico City (MMC) children have lower CSF Aβ1-42 and brain-derived neurotrophic factor (BDNF) concentrations versus controls (*p* = 0.005 and 0.02, respectively) [12]. In an earlier study by our group, we applied the Montreal Cognitive Assessment (MoCA) to 517 MMC residents, aged 21.60 ± 5.88 y, with 13.7 ± 1.3 y of formal education [13]. The average MoCA score was 23.9 ± 2.8 (normal 26–30), and 24.7% and 30.3% scored ≤24 and ≤22, corresponding to mild cognitive impairment MCI ≤ 24 and dementia scores D ≤ 22 [13]. Thus, urban Mexican populations may be at a heightened risk of long-term neurodegenerative sequel, including dementia associated with high exposure to neurotoxic air pollutants [14,15,16,17,18].

It is in this scenario that urban Mexican women are at high risk of traumatic events and the prevalence of post-traumatic stress disorder (PTSD) is high, with a significant lack of social and familial protection or support [19,20]. Rebeca Robles-Garcia and colleagues [19] analyzed PTSD in urban Mexican women, with a special focus on biopsychosocial risk factors, and discussed urbanization as a key factor due to the concentration of poverty, substance use, and crime. They argue and we agree—women are at a greater social and economic disadvantage resulting in them becoming victims of collective and domestic violence. Moreover, there is evidence at the international level that the female gender is associated with a 1.5 to 2-fold increase in the risk of post-traumatic stress disorder [21,22] as well as of specific sleep disorders [23], particularly REM sleep behavior disorder, which is the focus of the current study [24].

The stress problem is also documented in teenagers. As many as 80% of high school students in highly violent MMC areas have significant mental pathology associated with stress [25]. The impact of PTSD has recently been shown to extend far beyond its distressing psychological and social manifestations: there is a substantial body of evidence linking PTSD to a long-term risk of dementia, even after correction for potential sources of bias [26,27,28,29,30,31]. Olivé et al. [32] discuss the degeneration of the human basal forebrain as a putative link between persistent traumatic memory in PTSD and anterograde memory deficits in neurodegeneration, a plausible pathway in the setting of AD and PD pathology.

In this neurodegeneration setting, it is of interest that idiopathic REM sleep behavior disorder (iRBD), a parasomnia frequently associated with PD and dementia with Lewy Bodies [33], is more frequent in veterans with PTSD and traumatic brain injury (TBI + PTSD) [34], raising the possibility of a common denominator to iRBD and PTSD.

In light of the above information, this study examines two key aspects of psychological stress and sleep—namely PTSD and RBD in young middle-class Mexican adults residing in 101 Mexican cities. We hypothesized that, in this population, with documented quadruple abnormal protein brain aggregates—marking AD, PD, and TDP-43 pathology—and clinically by cognitive deficits, there would be a strong association between pRBD and the symptoms of PTSD [6,7,8,9,10,13]. 

We selected two extensively validated instruments: the Spanish version of the Impact of Event Scale–Revised (IES-R), which includes 22 items measuring three clusters of PTSD risk: intrusion, avoidance and hyperarousal symptoms [35,36,37,38,39,40,41] and the REM Sleep Behavior Disorder Single-Question Screen (RBD1Q) to assess RBD [42].

## 2. Methodology

The current study was conducted in 101 urban areas across Mexico ranging in population from ~21.8 million people, i.e., MMC, to towns with ≤10,000 residents. The research was conducted according to the Revised Helsinki Declaration of 2000 and the study was approved by the Universidad del Valle de México ethical and research committees. The data collection period was from 2 June to 3 July 2020, an anonymous online survey platform was used, and subjects were invited to participate. The invitation was aimed at college educated adults (≥18 y), either taking university classes online and/or working, followed by an explanation of the nature and purpose of the survey. Once the user consented to participate in the survey, they completed the following questions:
Demographic information, including city of residency, age, sex, formal education years, weight and height. Residents in Metropolitan Mexico City (MMC) and residents across 100 urban areas in the country (non-MMC) were included.The Spanish version of the Impact of Event Scale–Revised (IES-R) was applied. For evaluation, the scores were divided into four categories: 0–23, no psychological impact or normal score; 24–32, mild psychological impact; 33–36, moderate psychological impact; ≥37, severe psychological impact. A score of 33 or higher was selected as the most appropriate cut-off value for significant symptoms of probable PTSD in keeping with the literature [36,37,38,39,40,41]. The IES-R has well established validity and reliability [41] and in this work, the Cronbach’s alpha was 0.832.The REM Sleep Behavior Disorder Single-Question Screen (RBD1Q) assessing dream enactment with a single yes or no response. This question has a sensitivity of 94% and a specificity of 87% and consistently detects RBD in elderly populations [42,43,44].


### Statistical Analysis

We calculated the descriptive statistics of all relevant variables in each group of MMC residents and non-residents, as well as in all participants combined. Next, we categorized the subjects in each residence-group according to their IES-R score. We also explored the summary statistics of numerous relevant variables within various groups and categories. We tested for equality of the subscale scores among MMC and non-MMC residents by performing independent-sample *t*-tests. We also tested for equality of the subscale scores within the category of IES-R ≤ 32, ≥33 and ≥37, among the subjects having and not having PTSD, pRBD or having neither. For categorical variables (such as sex), we used Pearson’s Chi-squared test. Finally, we calculated sample odds-ratios of the risk of PTSD and pRBD and calculated 99% confidence intervals of the corresponding true odds-ratios. We performed the statistical analyses using Excel and the statistical software “R” (http://www.r-project.org/, accessed on 1 May 2021).

## 3. Results

This is a cross sectional study of 4502 middle-class, college educated subjects with 61.48% women (Table 1). The mean total IES-R score was 33.30 ± 15.28 and 22.61% responded yes to the RBD1Q.

The majority of individuals (50.44%) scored above the cut-off value IES-R ≥ 33 for PTSD and no differences were seen in the number of individuals in the severe category (IES-R ≥ 37) between MMC versus non-MMC residents (*p* = 0.14, Chi-squared test) (Table 2).

An evaluation of the symptom subscales in the IES-R showed a significant difference in the responses to stress between MMC and non-MMC residents, with higher scores for intrusion (*p* = 0.0180) and hyperarousal (*p* = 0.03) in MMC, but non-significant statistical differences in avoidance (*p* = 0.67) (Table 3).

Strikingly, 29.77% of all IES-R ≥ 33 subjects answered yes to the pRBD question, with MMC residents reaching 32.71% (Table 4).

On analysis of the subgroup with severe PTSD symptoms (IES-R score of ≥37), those screening positive on the RBD1Q had significantly higher symptom scores on intrusion and hyperarousal *p* < 0.0001 (Table 5).

For the entire sample, women had higher odds of developing symptoms of PTSD (OR = 1.6562, 99% confidence interval of true OR [1.6400, 1.6725]) and pRBD (OR 1.1207 [1.1051, 1.1365]) than men. Significantly, for MMC residents, the odds-ratio of PTSD versus non-PTSD subjects developing pRBD was 2.6218 [2.5348, 2.7117]. Residing in non-MMC regions was also associated with a higher risk of pRBD in women (OR 1.2408 [1.2094, 1.2729]) and an increased risk of screening positive for PTSD compared to men (OR1.9137 [1.8812, 1.9467]). For the purpose of sample size justification and power calculations of two-sample *t*-tests and two-sample proportion tests, Table 6 and Table 7 provide numerical details. For example, to detect an effect size of 0.10 with 80% power we need about 1570 subjects in each group in comparison. This justifies the sample size in our data.

## 4. Discussion

We open the discussion by emphasizing that this study explores a group of young, educated, middle-class and mostly female Mexican subjects, with significant published documentation of AD, PD and TDP-43 pathology, using forensic autopsy cases and extensive published cognitive, olfaction, gait and equilibrium and brainstem auditory evoked potentials (BAEPs) studies in matching populations—a situation that makes this study very different from the average PTSD and sleep behavior disorder cases in the literature: the average case focuses on individuals who are male, significantly older, of low socioeconomic status, and possess extensive co-morbidities [6,7,8,9,10,12,13,26,27,28,29,30,31,32,33,34,42,43,44,45,46,47]. 

Moderate to severe psychological distress as measured by the IES-R was found to affect more than 50% of individuals and those with scores ≥ 33—suggestive in the literature of a probable diagnosis of PTSD [38,39]—showed a strong association with pRBD. Remarkably, for MMC residents, having moderate to severe psychological stress increased their odds of having pRBD by 2.6-fold.

The association between stress and sleep related disorders and neurodegeneration is the core of this study. Song et al., explored a Swedish population of 44,839 individuals with stress-related disorders and their 78,482 unaffected siblings with a mean age of 47 years at the beginning of the 4.5 years (2.1 to 9.8 y) follow-up [26]. Compared with their control siblings, the stress diagnosed individuals had a higher risk of developing a vascular (HR 1.80; 95% CI 1.40–2.31) neurodegenerative disease versus Alzheimer’s (HR1.36; 95% CI 1.12–1.67) [26]. Desmarais and colleagues [27] using keywords in their meta-analysis: PTSD and dementia found 25 articles reporting patients aged 67.4 ± 7.1 y with PTSD developing dementia (n = 14) and new onset or worsening of PTSD in patients 80.3 ± 11.1 y with dementia (n = 11). The authors discussed how PTSD could represent a prodromal stage of dementia. Interestingly, in their review work, Alzheimer’s disease and vascular dementia were the most common neurodegenerative conditions, as were subjects with a history of early-life trauma. The authors concluded that repeated acute stress events and chronic stress conditions likely play a role in the bidirectional PTSD and dementia (i.e., neurodegeneration) relationships [27].

Of great interest to our study, are the clinicopathological correlations of 172 RBD cases by Boeve et al. [48]. The study group had 83% males, aged 20–93, and an average age at death of 75 ± 9 years. Their clinical diagnosis included Parkinsonism (n = 151), cognitive impairment (n = 147), and autonomic dysfunction (n = 42). At autopsy, Lewy body disease (LBD) (n = 77), combined LBD and AD (n = 59), multiple system atrophy (MSA) (n = 19), AD (n = 6) and progressive supranuclear palsy (n = 2) were diagnosed. In Boeve et al., RBD study, 94% of the subjects had documented synucleinopathies [48].

Clinical signs and cognition deficits are very useful to detect in early stages of neurodegeneration. McDade et al. [49] described RBD as associated with gait changes, while Shin et al. [50] made the association between RBD, cognitive decline and hyposmia.

Our results have to be discussed in light of the PTSD and RBD documented neuropathology, clinical associations in the literature and the various observations in our study that are worth highlighting. First, earlier research by our group has documented hallmarks of neurodegeneration in young ≤40 year MMC residents, as evidenced by: (a) Alzheimer, Parkinson and TDP-43 neuropathology and high concentrations of highly reactive, oxidative and magnetic metal-rich nanoparticles (NPs) including exogenous engineered Ti-rich nanorods and non-metal nanoparticles [6,7,10]. (b) Olfactory bulb AD and PD pathology, along with olfaction deficits, are reported in MMC children and young adults and remarkably, Apolipoprotein E allele 4 carriers (APOE4) are 4.57 times more likely to die by suicide versus APOE 3 carriers, and have higher chances of exhibiting olfactory bulb neuronal alpha-synuclein aggregates and hyperphosphorylated tau neurites and tangles [8,47]. (c) Significant cognition deficits, gait, equilibrium and BAEPs abnormalities are present in seemingly MMC healthy children and young adults [13,45,46].

Second, the current study found a high prevalence (22.6%) of pRBD in 4502 young people and this was strikingly higher (32.7%) in MMC residents with an IES-R score ≥ 33. These results are in sharp contrast to current literature, where most pRBD patients seeking medical care after injuries or at the request of partners, are male, with low education and over the age of 60 y [43,51,52].

Third, for subjects with severe psychological stress (IES-R ≥ 37), those screening positive for pRBD had significantly higher intrusion and hyperarousal responses (<0.0001) indicating their higher risk with psychiatric morbidity [53].

The complex interplay between stressors, neurocognitive and neuropsychiatric manifestations, and documented evidence of quadruple misfolded protein neurodegenerative pathologies affecting Mexican urbanites from earliest childhood, supports a plausible relationship between neuro-psychiatric-cognitive features and evolving neurodegeneration. Ubiquitous, airborne and environmental, metal-rich magnetic nanoparticles, including e-waste metals, are a common denominator in the brains and hearts of MMC residents [6,7,9,54,55]. Strikingly, significant concentrations of ferrimagnetic NPs from cerebellum > tectum/tegmentum/periaqueductal gray PAG > substantia nigra acquire great importance when one considers the role of the brainstem and cerebellum in emotions, affective behavior, autonomic output, sleep-wake cycles, posture, gait, and major cholinergic innervations of the thalamic relay nuclei and the thalamic reticular nucleus [6,56,57,58,59,60,61].

Brainstem and supratentorial neuropathology in young MMC residents are extensive and could be playing a crucial role in the development of PTSD and pRBD (Figure 1) [26,27,28,29,30,48,49,62,63].

We want to draw attention to the fact that most of the brainstem, and specifically the substantia nigrae pathology in young MMC subjects [6], is in the form of hyperphosphorylated tau, in sharp contrast with dominant synucleinopathies in Boeve’s work [48].

The main strengths of this study are the sample size of young, educated, middle-class, subjects across 101 Mexican cities, and the published data on neuropathology, CSF, cognition, brainstem auditory evoked potentials, olfaction and gait and equilibrium assessments in this target population. Our results strengthen the possibility of a robust connection between moderate to severe psychological stress and RBD and the association with neurodegeneration in young adults. Moreover, the key brainstem neuropathology marker in young Mexican urbanites is hyperphosphorylated tau.

We are aware of the limitations—including the screening for pRBD with the REM Sleep Behavior Disorder Single-Question instrument—requires that even assuming high sensitivity, a differential parasomnias diagnosis and other disorders, i.e., obstructive sleep apnea, has to be ruled out, and the gold standard diagnosis is an overnight video polysomnography (PSG) [34,43,44]. The need to implement a PSG in a representative sample is clear and urgent. The IES-R cut-off scores for a preliminary PTSD diagnosis are supported in the literature [38,39]. More than 50% of the subjects in this study had scores ≥33 reflecting a probable diagnosis of PTSD, however, there is a need to confirm the PTSD diagnosis using a diagnostic interview for PTSD—the DSM-5 PCL (PTSD Checklist; PCL)—and to address the overlap and boundaries between ICD-11 definitions of complex post-traumatic stress disorder (C-PTSD) and personality disorder [64,65,66,67].

Finally, ethnic minorities are under-represented in PTSD and RBD studies and clinicians should be aware there are striking ethnic differences in rates of adult and childhood trauma exposure and PTSD among minorities [68].

## 5. Conclusions

We argue pRBD and PTSD in the third and fourth decades of life are not isolated diagnoses. Rather, they are likely to be part of a continuum of neurocognitive, neuropsychiatric, gait, equilibrium, and auditory manifestations starting in childhood and reflecting ongoing quadruple misfolded proteins. We can’t ignore the neuropathology findings, 99.5% of 203 consecutive forensic autopsies in MMC could be staged for AD and the main marker was p-τ across all cases. It is concerning that the presence of pRBD in a large PD population-based study [69] is a clinical marker for faster cognitive decline.

We are aware of the overlap between neurodegenerative diseases. Most cases of dementia will have evidence of AD, vascular changes, PD, and TDP-43 pathology as shown by Karanth et al. [70]. Karanth and collaborators concluded that “quadruple misfolded proteins appear to be a common substrate for cognitive impairment and to be associated with an aggressive course of disease that typically ends with severe dementia” [70]. They also added that the overlap of aberrant proteins may complicate efforts to identify therapies to treat and prevent Alzheimer’s disease. We agree with them.

Our previous work focused on the relationship between air pollution and early neuropathological changes suggestive of dementia, which has been confirmed in systematic reviews of the literature [71]. In the current study, we have chosen to focus on the link between post-traumatic stress disorder and REM sleep behavior disorder, because the emerging evidence of a significant increase in the risk of subsequent neurodegeneration in patients with PTSD [72] and that disturbed sleep may be a significant mediator of this association [73]. There is also evidence that REM sleep behavior disorder, which has been consistently associated with neurodegenerative disorders, is more frequent in individuals with PTSD [34,74]. Potential cellular mechanisms underlying this association, such as oxidative stress and inflammation, are being elucidated [75], and these mechanisms overlap substantially with those that link environmental pollution and neurodegeneration [76]. Therefore, we wish to highlight the potential of a synergistic interaction between these two factors in increasing the subsequent risk of neurodegenerative disorders, particularly dementia. Though not directly examined in this study, this hypothesis is supported by an extensive body of literature.

We strongly support that we need to understand the factors that contribute to the neurodegenerative overlapping pathogenic cascade, and studying young populations at risk for neurodegeneration will enable prevention, earlier detection, and the discovery of targeted strategies to stop the development and progression of fatal neurodegenerative diseases affecting our populations. It is a matter of significant concern that women in this cohort were particularly affected by PTSD and pRBD. It is critical that these women are identified, diagnosed and treated, chiefly because of their higher psychiatric morbidity risk [53] and the fact that urban Mexican women are at very high risk for collective and domestic violence [19].

Understanding the interplay between neurodegeneration, sleep disorders, and PTSD in the first four decades of life ought to be a research priority for one key reason: identifying individuals at the earliest disease stages will shed light on molecular pathways that could be targeted for interventions aimed to avoid or ameliorate neurodegeneration, decrease stress, and ultimately to develop early disease-modifying treatments for AD, PD and TDP-43 diseases.

## Figures and Tables

**Figure 1 ijerph-18-06689-f001:**
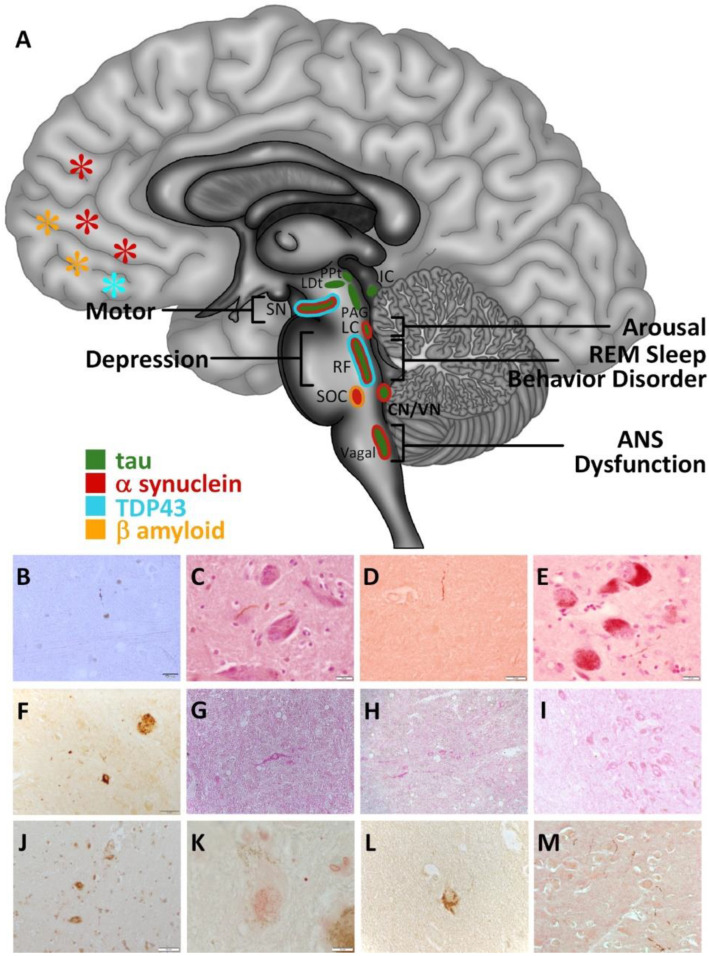
Distribution of misfolded proteins and neuropathological hallmarks in the brainstem and frontal cortex in Metropolitan Mexico City (MMC) individuals younger than 40 years [6,7,8,9,10,45,46]. The brainstem is an early target (motor, depression, arousal dysfunction and Autonomic Nervous System (ANS) dysfunction) [56,57,58,59,60,61] of misfolded proteins in Alzheimer, Parkinson and transactive response DNA-binding protein TDP-43 pathology in children and young adult residents in MMC [6]. (**A**) Brainstem pathology associated with RBD includes lesions in the pre-coeruleus, the sub-latero-dorsal nucleus, magnocellular nucleus, the locus coeruleus, degeneration in pedunculopontine nucleus, and corticothalamic circuits affecting the pathways that regulate REM sleep in the brainstem. Autonomic nervous system dysfunction, motor alterations, depression, and arousal mechanisms are impaired when the brainstem is involved in patients with dementia with Lewy bodies, Parkinson’s disease, multiple system atrophy and Alzheimer’s disease. (**B**) Positive hyperphosphorylated tau neurite in the mesencephalic reticular formation in an 11 m old baby (PHF-tau 8, Innogenetics, Belgium). (**C**) Substantia nigra pars compacta in a 17 year old male with + tau neurites (PHF-tau 8, brown DAB, counterstained with hematoxylin). (**D**) Pedunculopontine nucleus positive tau neurite in a 12 y old boy (PHF-tau 8, brown DAB). (**E**) Locus coeruleus neurons with + tau neurites in a 37 year old female (PHF-tau 8, brown DAB). (**F**) Midbrain, substantia nigra + tau tangles in neurons + neurites and plaques in a 40 year old male (PHF-tau 8, brown DAB). (**G**) Three-year-old boy + α-Syn neuron in the region of the medial lemniscus in the lower pons (α-Syn phosphorylated at Ser-129, LB 509, InVitrogen, Carlsbad, CA) (Red product). (**H**) Eleven y old female, lower medulla with numerous α-Syn positive neurons and neurites in the reticular formation region (LB509, red product). (**I**) Thirteen-year-old girl, substantia nigra pars compacta neurons + α-Syn (LB509, red product). (**J**) Fourteen-year-old male, pontine reticular formation several neurons with positive TDP-43 staining around mostly clear nuclei (TDP-43, mab 2G10, Roboscreen GmbH, Leipzig, Germany) (DAB, brown product). (**K**) Twenty-seven-year-old male, substantia nigra pars compacta neuron with neuromelanin degranulation and clearing of the nucleus (TDP-43, mab 2G10) (red product). (**L**) Frontal neuron tau positive in a 13-year-old female (PHF-tau 8, DAB brown product). (**M**) Eighteen-year-old male, frontal tau neurites cluster and neurons with Aβ perinuclear accumulation (PHF-tau 8, DAB brown product and beta amyloid 17–24 4G8 Covance, Emeryville, CA, USA, red product).

**Table 1 ijerph-18-06689-t001:** Demographics in the 4502 participants: all subjects, Metropolitan Mexico City, and non-Metropolitan Mexico City subjects. Data in Mean (SD).

Categories	All Subjects n = 4502	MMC n = 1865	Non-MMC Subjects n = 2637
Age	29.31 (10.31)	30.88 (10.83)	28.20 (9.77)
Sex (M/F)	1734/2768	731/1134	1003/1634
Years Education	15.76 (2.69)	15.91 (2.77)	15.66 (2.63)
BMI	25.56 (5.10)	25.51 (5.01)	25.59 (5.17)
IES-R total score	33.30 (15.28)	33.73 (15.46)	32.99 (15.15)
RBD1Q	Y/N = 1018/3484 (22.61%)	Y/N = 455/14,010 (24.40%)	Y/N = 563/2074 (21.35%)

**Table 2 ijerph-18-06689-t002:** IES-R results in the four categories: normal, mild, moderate, and severe psychological stress. 4502 participants: all subjects, MMC and non-MMC subjects.

Categories IES-R	All Subjects n = 4502	MMC n = 1865	Non-MMC Subjects n = 2637
Normal (0–23)	1215 (26.99%)	515 (27.61%)	700 (26.54%)
Mild (24–32)	1016 (22.57%)	393 (21.07%)	623 (23.63%)
Moderate (33–36)	467 (10.37%)	185 (9.92%)	282 (10.69%)
Severe (≥37)	1804 (40.07%)	772 (41.39%)	1032 (39.14%)

**Table 3 ijerph-18-06689-t003:** IES-R subscale results in all, MMC, and non-MMC. Mean (SD).

Subscale Score	All	MMC	Non-MMC
Intrusion	11.2 ± 6.5	11.5 ± 6.6	11.0 ± 6.4
Avoidance	12.0 ± 6.6	12.0 ± 6.5	12.1 ± 6.6
Hyper-arousal	9.9 ± 5.6	10.1 ± 5.7	9.8 ± 5.5

**Table 4 ijerph-18-06689-t004:** Demographics for participants with IES-R scores of ≥33. n = 2271, all; n = 957, MMC; n = 1314 non-MMC subjects. Data in Mean (SD).

Variables IES-R ≥33	All = 2271/4502	MMC = 957/1865	Non-MMC = 1314/2637
Age	29.03 (9.78)	30.66 (10.43)	27.85 (9.11)
Sex (M/F)	741/1530	341/616	400/914
Years Education	15.74 (2.61)	15.82 (2.68)	15.68 (2.55)
BMI	25.59 (5.08)	25.56 (5.05)	25.61 (5.11)
IES-R total score	45.46 (9.92)	45.90 (10.01)	45.13 (9.85)
RBD1Q	Y/N = 676/1595 (29.8%)	Y/N = 313/644 (32.7%)	Y/N = 363/951 (27.6%)

**Table 5 ijerph-18-06689-t005:** Sub-scores in intrusion, avoidance and hyperarousal in participants who answered yes and no on the RBD1Q with total IES-R scores ≥37.

IES-R Total Score	Intrusion	Avoidance	Hyperarousal
≥37, RBD1Q YES	18.1 (5.1)	16.5 (5.7)	16.5 (4.5)
≥37, RBD1Q NO	16.1 (4.7)	16.9 (5.2)	13.9 (4.3)
*p*-value	<0.0001	0.1773	<0.0001

**Table 6 ijerph-18-06689-t006:** This table shows the minimum sample size required in each group to achieve the given power for detecting the respective effect size when the level of significance is α = 0.05 in all cases. This is for a two-sided two-sample *t*-test when the population standard deviation in both groups is roughly the same.

Effect Size = (µ_1_ − µ_2_)/σ	Sample Size in Each Group (Power = 0.70)	Sample Size in Each Group (Power = 0.80)	Sample Size in Each Group (Power = 0.90)
0.05	4938	6280	8406
0.10	1235	1570	2102
0.15	549	698	934
0.20	309	393	526
0.25	198	252	337
0.30	138	175	234

**Table 7 ijerph-18-06689-t007:** This table shows the minimum sample size required in each group to achieve the given power for detecting the inequality in population proportions when the level of significance is α = 0.05 in all cases. This is for a two-sided two-sample proportion test.

p_1_	p_2_	Sample Size in Each Group (Power = 0.70)	Sample Size in Each Group (Power = 0.80)	Sample Size in Each Group (Power = 0.90)
0.10	0.15	540	686	918
0.10	0.20	157	199	266
0.10	0.25	79	100	133
0.20	0.25	861	1094	1464
0.20	0.30	231	294	392
0.20	0.35	109	138	185
0.30	0.35	1083	1377	1842
0.30	0.40	281	356	477
0.30	0.45	128	163	217

## Data Availability

All data are available in the paper.

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
