# Peer review of "Brainstem Quadruple Aberrant Hyperphosphorylated Tau, Beta-Amyloid, Alpha-Synuclein and TDP-43 Pathology, Stress and Sleep Behavior Disorders"

_ijerph, 2021, doi:10.3390/ijerph18136689_

Round 1
Reviewer 1 Report
In the present study, Lilian Calderón-Garcidueñas and colleagues aimed to identify an association between Post-Traumatic Stress disorder (PTSD) and potential REM sleep behavior disorder (pRBD) in Mexico. The authors concluded that it is documented an association between PTSD and pRBD, strengthening the possibility of a connection with misfolded proteinopathies in young urbanites. Moreover, Mexican young women are at high risk for stress and sleep disorders.
Overall, I think that the paper is interesting, and it could be also of interest to the readers of International Journal of Environmental Research and Public Health.
I raise a series of specific points to address carefully for improve the quality of paper.
1) The authors should justify the sample size and need to do a power calculation.
2) How can you explain that young women from a pathophysiological point of view are at high risk for stress and sleep disorders? This topic is crucial to address to evaluate the role of environment and associated behavior in terms of lifestyle and public health.
3) Do you identify different risk factors to explain the association between PTSD and pRBD? Please also justify this aspect and kindly insert appropriate references.
4) Which is the role of Environmental Toxins in this context? Please kindly add two/three sentences and please insert appropriate references in the discussion section of paper focusing this key point.
5) In light of the results here obtained, please to discuss on the possible application of nutraceutics and/or antioxidants/antinflammatory compounds, that could minimize the negative effects of Post-Traumatic Stress disorder (PTSD) in humans.
Author Response
Enclosed answers to reviewer #1

Reviewer 2 Report
Calderón-Garcidueñas and colleagues describe a study that tests for a potential association between PTSD and RBD in subjects from metropolitan areas vs. what I believe they have in mind as rural areas. The study might be of interest to both the lay (e.g., the general audience) and clinical community, however, as presented, it is disorganized and well presented, with some of the authors' conclusions being unsubstantiated based on the presented data.
- The authors should have been clearer about their terminology. For example, what is non-MMC? Are these rural areas? Similarly, is RBD a measure of sleep disorder and if so, why choosing this measure?
- The manuscript would have benefitted from simplifying. The authors could (or should have) just stop at reporting the association between the suggestive scores of PTSD and RBD without involving neurodegeneration.
- Connecting the results from their study with neurodegenerative diseases is "a leap of fate" rather than a rigorously tested hypothesis. Even if they want to do that, a lot more information is need from what they have reported in their study.
- The effect sizes should have been reported as well.
Author Response
Answers to reviewer #2

Round 2
Reviewer 1 Report
The authors responded sufficiently to requests and improved the manuscript accordingly. Thank you for your revisions.
Reviewer 2 Report
The authors have addressed all of my comments